# RSVP: Customer Intent Detection via Agent Response Contrastive and Generative Pre-Training

**Yu-Chien Tang, Wei-Yao Wang, An-Zi Yen, Wen-Chih Peng**

Department of Computer Science, National Yang Ming Chiao Tung University, Taiwan

tommytyc.cs10@nycu.edu.tw, sf1638.cs05@nctu.edu.tw,
azyen@nycu.edu.tw, wcpengcs@nycu.edu.tw

## Abstract

The dialogue systems in customer services have been developed with neural models to provide users with precise answers and round-the-clock support in task-oriented conversations by detecting customer intents based on their utterances. Existing intent detection approaches have highly relied on adaptively pre-training language models with large-scale datasets, yet the predominant cost of data collection may hinder their superiority. In addition, they neglect the information within the conversational responses of the agents, which have a lower collection cost, but are significant to customer intent as agents must tailor their replies based on the customers' intent. In this paper, we propose RSVP, a self-supervised framework dedicated to task-oriented dialogues, which utilizes agent responses for pre-training in a two-stage manner. Specifically, we introduce two pre-training tasks to incorporate the relations of utterance-response pairs: 1) **Response Retrieval** by selecting a correct response from a batch of candidates, and 2) **Response Generation** by mimicking agents to generate the response to a given utterance. Our benchmark results for two real-world customer service datasets show that RSVP significantly outperforms the state-of-the-art baselines by 4.95% for accuracy, 3.4% for MRR@3, and 2.75% for MRR@5 on average. Extensive case studies are investigated to show the validity of incorporating agent responses into the pre-training stage[1].

## 1 Introduction

Task-oriented dialogue systems aim to assist users in finishing specific tasks, and have been deployed in a wide range of applications, e.g., tour guides (Budzianowski et al., 2018), medical applications (Levy and Wang, 2020), and especially in the e-commerce domain such as booking flights and hotels (El Asri et al., 2017). With the rapidly expanding market and customers' growing demands

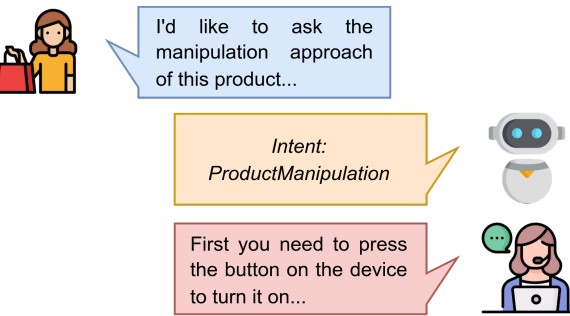

Figure 1: An example of the customer utterance (blue), the customer intent (orange), and the agent response (red).

for high-quality shopping experiences, providing precise answers efficiently through customer services has become increasingly vital for companies. Recently, companies, especially e-commerce companies, are developing their own AI models (Tao et al., 2018; Chen et al., 2019; Song et al., 2022) to automate repetitive tasks and reduce the burden on agents. The common approach is to frame this problem as an intent detection task (Liu et al., 2019a), which involves classifying a customer's utterance into one of the pre-defined intents.

Previous methods tackling intent detection tasks with deep neural models (E et al., 2019; Zhang et al., 2019) are able to learn semantic representation from customer utterances. However, these methods rely on large-scale high-quality labeled data requiring expensive and time-consuming annotations (Abro et al., 2022). Recent studies on intent detection primarily focus on pre-training language models with public dialogue datasets and fine-tuning them for downstream intent detection tasks (Casanueva et al., 2020; Mehri et al., 2020; Mehri and Eric, 2021; Zhang et al., 2021a). These methods have proven to be effective, but they heavily depend on the availability of additional public pre-training datasets, where large amounts of unlabeled data, especially when the language is not

---

[1] Code is available at https://github.com/tommytyc/RSVP.

in English, can be hard to acquire and collect due to the high cost (Liang et al., 2021). Furthermore, most of these studies only utilize customer utterances to detect intent, while neglecting the usefulness of metadata (e.g., agent responses) from in-house customer services systems, which can be efficiently collected with customer utterances and without the requirement of annotation cost. However, how to effectively utilize the agent responses to benefit intent detection is an unknown yet challenging problem, as the agent responses cannot be obtained before detecting the intent of customer requests during real-time deployment.

In light of this, we aim to incorporate agent responses with customer utterances in the training set to produce fine-grained knowledge. We hypothesize that if the model is able to provide an accurate response for a given utterance, it must have inferred the hidden intent behind the utterance. As shown in Figure 1, a customer may raise a question about the manipulation of a product. The agent response shows the concrete solution, *"Press the button on the device"*, to the customer utterance due to the agent's understanding of the intent being the manipulation approach to the product. If the pre-trained model understands that the agent's response will include the manipulation approach to the product, the intent detection model will be able to classify the utterance into the intent: *ProductManipulation*.

In this work, we propose RSVP, a two-stage framework composed of a pre-training step and a fine-tuning step to utilize the agent responses to effectively learn the intent of customer utterances by introducing two pre-training tasks to learn a much more reliable conversational text encoder which has been further pre-trained to predict an accurate response. The first task, *Response Retrieval*, follows a dual encoder paradigm and models the customer utterance as a dense retrieval to a batch of agent responses. This reformulation enables the pre-trained language model to discriminate between the correct response and the others. The second task, *Response Generation*, constructs a conversational question-answering problem with the utterances and responses. By directly learning to answer the customer utterances, the pre-trained language model can improve the extracted semantic knowledge of the agent responses. Finally, the pre-trained language model will be used to fine-tune the intent detection task. As the pre-training tasks have aligned the model better with the agent

responses, we expect that the adapted model can infer the intent better for each incoming utterance.

Our contributions can be summarized as follows:

- We propose a novel intent detection approach for customer service by integrating the agent response as the pre-training objectives. Our method is flexible for the intent detection problem and can also be deployed to a variety of domains with task-oriented dialogue systems in similar contexts.

- We designed two pre-training tasks, namely *Response Retrieval* and *Response Generation*, that enable the language model to learn the implicit intent of customer utterances with agent responses and benefit from the recent advancements in contrastive learning.

- Experimental results on customer intent detection benchmarks demonstrate the effectiveness of our methods without the need for additional pre-training datasets.

## 2 Related work

Existing intent detection approaches have shown promising results by leveraging the ability of transfer learning with two-stage pre-training and fine-tuning process. Specifically, these works (Henderson et al., 2020; Mehri et al., 2020; Casanueva et al., 2020; Mehri and Eric, 2021) involve pre-training models in a self-supervised manner with mask language modeling (MLM) on large public dialogue datasets, and fine-tuning them on the target intent datasets. Zhang et al. (2021a) presented a continued pre-training framework, where they pre-trained the model with MLM and cross-entropy losses from a few publicly available intent datasets to improve the few-shot performance on cross-domain intent detection tasks.

Additionally, a line of studies has introduced pre-training methods incorporating a variety of relevant NLP tasks, such as natural language inference (NLI), to leverage knowledge from other public datasets. Zhang et al. (2020) recast intent detection as a sentence similarity task and pre-trained the language model on NLI datasets with BERT (Devlin et al., 2019) pairwise encoding, and they came up with a nearest-neighbor framework to take advantage of the transfer learning. However, their system is computationally expensive due to the full utilization of training data in

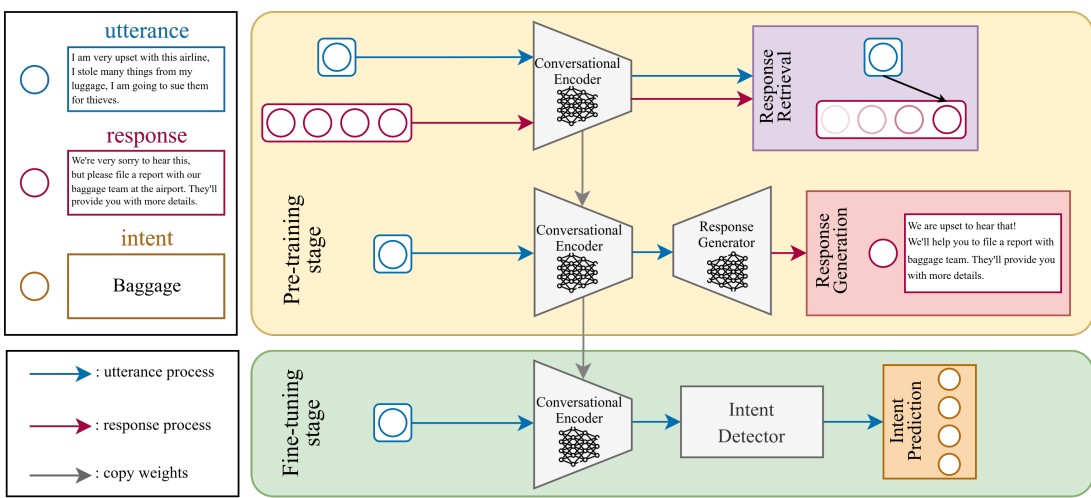

Figure 2: The overall framework of RSVP with two related pre-training tasks: **Response Retrieval** and **Response Generation** to explore the agent responses in the dialogue system and learn the customer intents.

both the training and inference stages. Zhang et al. (2021b) pre-trained a language model on intent detection datasets with self-supervised learning and fine-tuned it with supervised-contrastive learning (Gunel et al., 2021). Zhang et al. (2022) designed an additional regularization loss term to tackle the anisotropy problem, as they found that supervised pre-training may yield a feature space where semantic vectors fall into a narrow cone. Yehudai et al. (2023) reformulated intent detection as a question-answering task and used intent label names as the answer to the customer utterances. Our method shares some similarities with Vulić et al. (2021), who proposed pre-training the language model on a public Reddit dataset (Henderson et al., 2019) with a response ranking task, and then fine-tuned it on an intent detection task by contrastive learning. In contrast, in order to take the customer service agent's response into consideration and relieve the large-scale data labeling effort, we pre-train directly on the customer service dialogue with our two pre-training tasks: Response retrieval, which introduces a response selection task by retrieval-based batch contrastive loss, and response generation, which formulates mimicking agent responses by a text generation task.

## 3 Method

### 3.1 Preliminary

**Problem definition.** Our method aims to build an intent detector from an annotated dataset $\mathcal{D} = \{(u_1, y_1), (u_2, y_2), ..., (u_{|\mathcal{D}|}, y_{|\mathcal{D}|})\}$, where $y_i$ is the intent label of utterance $u_i = [u_i^1, u_i^2, ..., u_i^H]$

of length $H$, and $y_i$ is from one of the pre-defined customer intents $C$. The goal is to classify the correct intent label for the corresponding utterance. As described in Section 1, we denote the agent response $r_i = [r_i^1, r_i^2, ..., r_i^T]$ of length $T$ for each $u_i$ to incorporate the metadata in the model.

**RSVP overview.** We introduce a simple yet general framework RSVP with two pre-training tasks from the agent responses, which aims to mimic how the agents think when they encounter an incoming utterance and try to respond to it. The first pre-training task, *Response Retrieval*, is proposed to match the relations between customer utterances and agent responses (Section 3.2). The second pre-training task, *Response Generation*, is designed to learn the structured contexts of the agent responses (Section 3.3). In these manners, the model is able to provide appropriate responses to utterances and has accordingly aligned with the thoughts of real agents. Afterwards, we employ this pre-trained model in the intent detection fine-tuning stage and train the classification task with intent labels of each utterance. Besides, to leverage the recent success of consistency regularization (Verma et al., 2019), which has been proven to be effective in improving classification performance, we introduce an additional contrastive loss term to enhance its robustness (Section 3.4).

### 3.2 Pre-Training: Response Retrieval

The response retrieval task is to accurately retrieve the appropriate agent response for a given customer utterance by formulating it as a question-answering (QA) dataset. The advantages of applying the pre-

training task are two-fold: 1) pre-training with agent responses does not require additional annotation or a generic dataset as the existing work does (Wu et al., 2020; Vulić et al., 2021), and 2) the structure of the utterance-response pair, in which the response aims to provide a related answer to the utterance and should thus share a similar representation (Karpukhin et al., 2020), enables a context-aware comprehension of the conversation. Specifically, we retrieve the utterance embedding $q_i \in \mathbb{R}^d$ and response embedding $p_i \in \mathbb{R}^d$, where $d$ denotes the vector dimension, with a shared conversational encoder $\phi(\cdot)$, composed of a BERT-based encoder and a linear layer. We adopt a pooling layer on top of the BERT-based encoder outputs to only consider the [CLS] token hidden representations and feed them to the linear layer with a tanh activation function. The process can be briefly denoted as:

$$q_i = \phi(u_i), \tag{1}$$

$$p_i = \phi(r_i). \tag{2}$$

To reinforce the model to learn the embedding space where the utterances and their corresponding responses are more likely to have stronger semantic relatedness as compared to irrelevant ones, we utilize an in-batch contrastive loss for each pair $q_i$ and corresponding $p_i$ as:

$$\mathcal{L}_{retr} = -\frac{1}{n}\sum_{i=1}^{n} log(\frac{e^{(sim(q_i,p_i)/\tau)}}{\sum_{j=1}^{n} e^{(sim(q_i,p_j)/\tau)}}), \tag{3}$$

where $n$ is the batch size, $\tau$ is a temperature parameter, and $sim(\cdot, \cdot)$ denotes the cosine similarity between two given embeddings. The corresponding response of an utterance is viewed as the positive sample and the other responses in the same batch as the negative samples.

### 3.3 Pre-Training: Response Generation

The goal of *Response Generation* is to ask the model to generate an appropriate response based on the corresponding utterance. We hypothesize that the model is able to precisely classify the intent if it has the capability to respond to an utterance, as the agents also have to recognize the intent before they respond. It is worth noting that our primary focus is not on the quality of the generated responses but rather on the capability of the model to capture the underlying intents of the utterances.

This sequence-to-sequence task can be achieved with an encoder-decoder architecture. While T5 (Raffel et al., 2020) is much more common as an encoder-decoder architecture, we make the hypothesis that the BERT encoder has been pre-trained to do classification tasks with its hidden representation and can accordingly perform better in the downstream intent detection task, yet T5 is pre-trained to directly generate text. Therefore, the pre-trained conversational encoder is adopted as the encoder, and a pre-trained BERT is used as the decoder. There are two adjustments being made for the decoder initialization with BERT. First, the self-attention mechanism is masked to make the model only focus on the left side of the context. Second, an encoder-decoder cross-attention layer will be added to the decoder. Formally, for each utterance $u_i$ and response $r_i$, the model learns to estimate the conditional likelihood $P(r_i|u_i)$ via minimizing the cross-entropy loss:

$$\mathcal{L}_{gen} = -logP(r_i|u_i),$$
$$= -\sum_{t=1}^{T} logP(r_i^t|r_i^{1:t-1}, u_i). \tag{4}$$

We note that both the encoder and decoder are jointly updated to make the conversational encoder learn how to respond to an incoming utterance. In this way, the encoder can simulate the real-world scenario where a new agent is learning to come up with an appropriate response and can thus further learn to align with the style of agents' responses.

### 3.4 Fine-Tuning: Intent Detection

We employ a Multi-Layer Perceptron (MLP) as the intent detector on top of the pre-trained conversational encoder. In this stage, the training utterances are the same as in the pre-training stage, and we exclude the responses during fine-tuning since the customer's intent should be detected before agents start to respond in the real world. Therefore, we discard the decoder and adapt the encoder in the fine-tuning stage. Specifically, the pre-trained conversational encoder is employed to retrieve the utterance sentence embedding $q_i$. We predict the estimated probability $\rho_i \in \mathbb{R}^{|C|}$ of the intents by the softmax output of the MLP classifier as follows:

$$\rho_i = \text{softmax}(\psi(q_i)), \tag{5}$$

where $\psi(\cdot)$ denotes an MLP.

RSVP aims to optimize the cross-entropy loss:

$$\mathcal{L}_{ce} = -\frac{1}{n}\sum_{i=1}^{n} y_i \cdot \log(\rho_i). \tag{6}$$

However, only incorporating cross-entropy loss may be sensitive to real-world noises (e.g., misspellings in the text or label noise) for performance

degradation (Ghosh and Lan, 2021; Cooper Stickland et al., 2023). In light of this, we integrate unsupervised learning with the cross-entropy loss to ensure the utterance embeddings remain consistent with their corresponding augmentations to further enhance the robustness. Inspired by the success of the contrastive learning (Gao et al., 2021), we adopt an augmented function $z$ against $q_i$ twice to get two views $\hat{q}_i = z(q_i)$ and $\bar{q}_i = z(q_i)$ as a positive pair. Specifically, we choose a dropout (Hinton et al., 2012) as $z$ since the dropout technique, which performs an implicit ensemble of different sub-models by simply dropping a certain proportion of hidden representations, has been proven to be effective as a regularization method (Wu et al., 2021; Liu et al., 2021). Combing it together with contrastive learning can further force $\hat{q}_i$ and $\bar{q}_i$ to be consistent with each other and have similar representations.

The unsupervised objective can be defined as:

$$\mathcal{L}_{uns\_cl} = -\frac{1}{n} \sum_{i=1}^{n} log(\frac{e^{(sim(\hat{q}_i, \bar{q}_i)/\tau)}}{\sum_{j=1}^{n} e^{(sim(\hat{q}_i, \bar{q}_j)/\tau)}}). \quad (7)$$

Finally, the objective function of RSVP is:

$$\mathcal{L}_{ft} = \mathcal{L}_{ce} + \lambda \mathcal{L}_{uns\_cl}, \quad (8)$$

where $\lambda$ is a weight hyper-parameter and is discussed in Appendix A.

# 4 Experiments

## 4.1 Intent Detection Datasets with Responses

Most widely studied public intent detection benchmarks (e.g., Clinc150 (Larson et al., 2019), Banking77 (Casanueva et al., 2020), and HWU64 (Liu et al., 2019b)) are not composed of user utterances and agent responses, which serve as an essential factor in task-oriented dialogue services. In order to evaluate our proposed RSVP in scenarios as discussed in the Section 1, we collected real-world Chinese datasets from **KKday**[2] with 50,276 samples, and adopted a public English dataset with 491 samples, **TwACS** (Perkins and Yang, 2019).

The KKday dataset annotated by 8 agents contains anonymous utterance-response pairs for the e-commerce travel domain. We removed the URL and Emoji from the utterances and responses to preprocess the dataset. We note that the dialogue in KKday consists of multi-turn utterances and responses; therefore, we concatenated all customer utterances and all agent responses respectively in

each dialogue into a single utterance and response for experiments. TwACS is composed of conversations between customer service agents from different airline companies and their customers on various topics discussed in Twitter posts. We used the processed TwACS[3] for our experiments. All results are averaged over 5 different random seeds.

## 4.2 Baselines

To evaluate the effectiveness of our proposed framework, we compare RSVP against multiple strong baseline approaches for intent detection tasks[4]: 1) **Classifier**: a BERT-based encoder which is fine-tuned with an MLP classification head and cross-entropy loss. 2) **ConvBERT** (Mehri et al., 2020): a BERT-based model pre-trained on a large unlabeled conversational corpus (Henderson et al., 2019) with around 700 million dialogues. 3) **DNNC** (Zhang et al., 2020): a discriminative nearest-neighbor model managing to find the best-matched sample from the training data through similarity matching. It pre-trains the language model with different labeled sources for NLI (Bowman et al., 2015; Williams et al., 2018; Levesque, 2011; Wang et al., 2018) and uses the pre-trained model for downstream intent detection. 4) **CPFT** (Zhang et al., 2021b): a two-stage framework that pre-trained a BERT-based model with self-supervised objectives in the first stage and fine-tuned with supervised contrastive learning in the second stage.

## 4.3 Quantitative Results

We use accuracy as the main evaluation metric, following (Zhang et al., 2021b). In addition, we also consider the scores of Mean Reciprocal Rank (MRR)@{3, 5} to evaluate the ability of each model to predict correct intent with a better rank, which can be served as another suggestion for agents using the intent detection system in real-world scenarios. Table 1 summarizes the results on the two datasets. We find RSVP consistently outperforms baseline methods across two datasets in terms of all evaluation metrics. Quantitatively, RSVP surpasses the state-of-the-art model, CPFT, by 1.7% and 8.2% for accuracy on KKday and TwACS respectively. It also improves CPFT by 1.3% and 5.5% for MRR@3, and 1.2% and 4.3% for MRR@5 on both datasets. These demonstrate

---

[2]https://www.kkday.com/en-us

[3]https://github.com/asappresearch/dialog-intent-induction

[4]The implementation details and dataset statistics are described in Appendix A.

| Model | KKday | | | TwACS | | |
|---|---|---|---|---|---|---|
| | Acc | MRR@3 | MRR@5 | Acc | MRR@3 | MRR@5 |
| Classifier | 51.76 | 67.16 | 69.02 | 59.60 | 69.93 | 71.59 |
| $+\mathcal{L}_{uns\_cl}$ | 52.68 | 67.75 | 69.55 | 63.20 | 72.00 | 73.36 |
| ConvBERT (Mehri et al., 2020) | 47.88 | 61.96 | 63.74 | 57.20 | 67.93 | 69.50 |
| $+\mathcal{L}_{uns\_cl}$ | 47.91 | 61.77 | 63.85 | 58.40 | 67.80 | 69.04 |
| DNNC (Zhang et al., 2020) | 49.79 | 63.07 | 65.25 | 34.00 | 43.13 | 45.55 |
| CPFT (Zhang et al., 2021b) | 52.55 | 67.27 | 69.09 | 63.60 | 71.80 | 73.64 |
| RSVP | **53.42** | **68.15** | **69.92** | **68.80** | **75.73** | **76.83** |

Table 1: Results ($\times 100\%$) on two intent detection datasets. The highest results for each metric are indicated in **boldface**, while the second best are underlined. Our improvements over the second-best baseline are significant with $p < 0.05$ (p-values based on Pearson's $\chi^2$ test).

the effectiveness of incorporating the agent responses into the intent detection task without the need for external annotated pre-training datasets.

To delve into the impact of contrastive learning in the fine-tuning stage, we experiment the variants of Classifier and ConvBERT during training and add contrastive learning along with cross-entropy loss ($+\mathcal{L}_{uns\_cl}$) as RSVP. We note that DNNC and CPFT are excluded due to the incorporation of multiple losses; therefore, adding an additional loss requires carefully adjusting the hyper-parameters. It can be observed that the auxiliary unsupervised contrastive learning also increases Classifier accuracy by 5.1% on average, and ConvBERT slightly performs better on accuracy as well. Meanwhile, RSVP still outperforms these baselines equipped with contrastive learning, which showcases the strength of our model's capacity.

### 4.4 Results on Larger Datasets

To further illustrate the comparisons between RSVP and the baselines, we experiment with two additional datasets, MultiWOZ 2.2 (Zang et al., 2020) and SGD (Rastogi et al., 2020), which contain user utterances, agent responses, and user intents in a conversation. It is noted that these two datasets were designed to detect the active intent of each utterance rather than the whole dialogue as our settings. To provide fair comparisons as ours, we reformulate the customer intent in a dialogue as the set of active intent appearing in all of the utterances for these two datasets (i.e., multi-label classification task). The F1 score and accuracy are incorporated as the evaluation metrics and the results are shown in Table 2. We note that the accuracy metrics may not be intuitive in multi-label classification tasks, and we consider evaluating it more strictly with subset accuracy – the set of labels predicted for a sample (with a model predicting

| Model | MultiWOZ 2.2 | | SGD | |
|---|---|---|---|---|
| | F1 | Acc | F1 | Acc |
| CPFT | 96.73 | 84.50 | 65.91 | 32.46 |
| RSVP | **98.27** | **91.04** | **72.88** | **45.77** |

Table 2: Results ($\times 100\%$) on two multi-label intent detection datasets. Our improvements over the second-best baseline are significant with $p < 0.05$ (p-values based on Pearson's $\chi^2$ test).

probability > 0.5) must exactly match the corresponding set of labels in ground truth.

In terms of all datasets and metrics, we can observe that RSVP significantly outperforms the best-performing baselines, CPFT, which further showcases the robustness of our method across various dataset scenarios. The results indicate that supervised contrastive learning loss in the fine-tuning stage of CPFT may restrict them to a multi-label context due to the difficulty of learning from the positive and negative pairs for each multi-intent instance. In contrast, we use unsupervised contrastive learning with the dropout function for augmentation.

### 4.5 Ablation Studies

**Pre-training Task.** To understand the importance of each framework component and the main factors leading to our improvements, we conducted several ablation studies and summarize the results in Table 3. First, we analyzed the effects of the pre-training stage by removing two pre-training tasks: response retrieval and response generation, respectively. We can see that both tasks are necessary to achieve better performances across all datasets and metrics, in line with the fact that a task-adaptive pre-training stage reinforces the model with the flexibility to adjust its feature space to the target task for boosting the downstream performance (Gueta et al., 2023).

| Model | KKday | | | TwACS | | |
|---|---|---|---|---|---|---|
| | Acc | MRR@3 | MRR@5 | Acc | MRR@3 | MRR@5 |
| RSVP | **53.42** | **68.15** | **69.92** | **68.80** | **75.73** | **76.83** |
| w/o *Response Retrieval* | 53.33 | 68.04 | 69.86 | 57.60 | 65.87 | 67.57 |
| w/o *Response Generation* | 52.91 | 67.63 | 69.37 | 67.60 | 73.87 | 75.67 |
| reverse pre-training tasks order | 52.91 | 67.61 | 69.37 | 63.60 | 70.60 | 72.20 |
| w/o $\mathcal{L}_{uns\_cl}$ | 52.82 | 67.85 | 69.58 | 62.80 | 72.87 | 73.61 |
| replace BERT with mT5 | 52.25 | 67.40 | 69.04 | 46.00 | 55.60 | 57.66 |

Table 3: Results ($\times 100\%$) of ablative experiments.

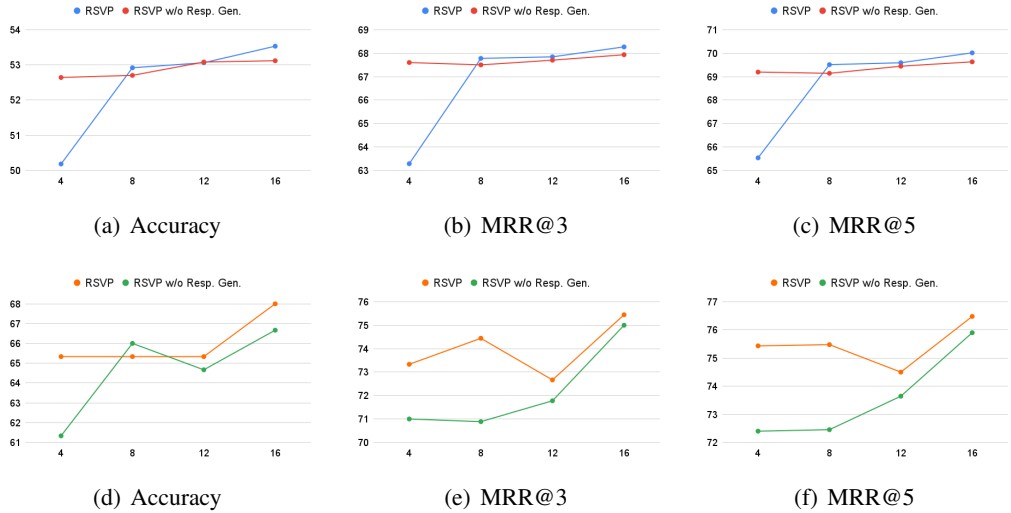

(a) Accuracy    (b) MRR@3    (c) MRR@5

(d) Accuracy    (e) MRR@3    (f) MRR@5

Figure 3: Effects ($\times 100\%$) of the batch size on the *Response Retrieval* training. (a-c) report the results on KKday with varying numbers of negative samples respectively. (d-f) report the corresponding results on the TwACS dataset.

It is worth noting that removing the response retrieval task results in an enormous drop in accuracy (-16.3%) on the TwACS. This is also consistent with the observation that a model with better utterance representation is essential when only a few examples are available (Choshen et al., 2022), given that TwACS is a relatively small dataset.

To investigate the effects of the pre-training order, we report the model variant which reverses the order of the two pre-training tasks, which demonstrates that the proposed order is superior to the reversed one. We hypothesize that the response generation task is more difficult than the response retrieval task, which is similar to the customer service scenario in which it is more difficult for a novice agent to directly come up with the appropriate response instead of picking one from a few candidates. This aligns with prior work showing that presenting training tasks ordered by difficulty level benefits not only humans but also AI models (Xu et al., 2020).

**Effects of** $\mathcal{L}_{uns\_cl}$**.** We also experiment with discarding the unsupervised contrastive learning loss in the fine-tuning stage and using only cross-entropy loss instead, which is a widely used setting in classification tasks. Our results show that the additional $\mathcal{L}_{uns\_cl}$ loss term leads to an enhancement in the accuracy of 5.7% on average, supporting the claim that contrastive learning is effective in terms of boosting the accuracy and robustness in the classification task.

**Effects of Encoder-decoder architecture.** We experiment with replacing the BERT encoder with the encoder part of mT5 (Xue et al., 2021) and using the whole mT5 in the response generation task, adopting `mt5-base` from (Wolf et al., 2020) for the two datasets. The conversational embeddings $q$ and $p$ are retrieved by the average encoder outputs across all corresponding input tokens, given that T5-based models do not have a [CLS] token. It can be observed that mT5 is not as effective as BERT across all datasets, especially on the TwACS with a drastic drop in accuracy (-49%). We make an assumption that the phenomenon comes from

| Utterances | Generated Responses | Predicted Intent |
|---|---|---|
| May I ask which SIM card you would recommend for my upcoming Europe trip? | Here is the information regarding Europe SIM card, which offers 4G high-speed internet access through eSIM. | WIFI/SIM Card inquiries |
| May I know when will the booking of this aquarium tickets on 10/9 be available? | Hello, you can visit the product page again in December to see if it is available for ordering (in January next year). | Date does not open yet |
| Could you confirm if my tickets have been successfully booked? The website shows that I used a coupon but no order was generated. | Hello, your order has been successfully placed. If you have any questions, please feel free to contact our customer service team at any time. Thank you. | Transaction failed |

Table 4: Examples of utterances, our generated responses, and predicted intent of KKday. All sentences have been translated into English.

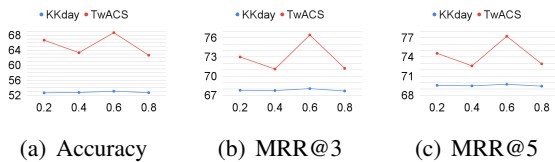

(a) Accuracy  (b) MRR@3  (c) MRR@5

Figure 4: Impacts of different $\lambda$ in the fine-tuning stage.

the need for a larger dataset to transform a larger model like mT5 into a conversational encoder and a response generator.

## 4.6 Parameters Analysis

**Number of negative examples in Response Retrieval.** To analyze the impacts of the number of negative examples in the response retrieval pre-training stage, we set batch size $n \in \{4, 8, 12, 16\}$ as summarized in Figure 3. We observe that increasing batch size leads to an enhancement in the performance in most cases. We ascribe the observed improvement to the increasing difficulty of the response retrieval task. By training the model to distinguish the correct response from a larger set of unrelated negative samples, the model is reinforced to learn more accurate representations of the utterances and responses. As a result, it gains enhanced capabilities to detect customer intents more effectively in the intent detection task.

**Impacts of $\mathcal{L}_{uns\_cl}$.** We also study the impact of $\lambda$ for unsupervised contrastive learning in the fine-tuning stage. We set $\lambda \in \{0.2, 0.4, 0.6, 0.8\}$ and report the results in Figure 4. RSVP performs stably with different numbers of $\lambda$, while the results show effects on TwACS. This is consistent with the observation of Zhang et al. (2021b) that classification performance may be sensitive to the weight of contrastive learning loss with limited training examples. Nonetheless, RSVP still performs better than all baselines in these cases.

## 4.7 Case Studies

**Analysis of generated responses and predicted intents.** We present a few randomly sampled utterances and their corresponding generated responses by the trained response generator with their predicted intents in Table 4. Although the quality of generated response is not the primary focus of this work, it can be seen that the generated response can effectively answer the utterance, which may open another direction to adopt this for automatic replies in the future. For instance, the model successfully recognizes the SIM Card recommendation intent of the utterance in the first example, tailors a reply with product information for the customer, and correctly predicts the customer intent: WIFI/SIM Card inquiries. This is supportive of our intuition that a model aligned with the thoughts of real agents can better understand the intent of customer utterances.

**Relations between intent boundaries and utterances.** In order to further understand the effect of the RSVP framework, we visualize the utterance representations from the test set of KKday extracted by different training stages of the conversational encoder: BERT, BERT after response retrieval (w/ Resp. Retr.), BERT after response generation (w/ Resp. Gen.), and BERT after full RSVP pre-training and fine-tuning (Full RSVP). Each sample point will be passed through a corresponding model and its high-dimension embeddings will be reduced to a two-dimension point with tSNE (van der Maaten and Hinton, 2012). We randomly chose 5 intent classes for the analysis as shown in Figure 5. All the trained conversational encoders exhibited clearer group boundaries compared to the vanilla BERT, indicating that the RSVP framework can extract better conversational representations for the downstream intent detection tasks. In addition, we notice that the *Price* intent cannot cluster as other intents in Figure 5 (b) and (c). The reason may be that the variety of utterances asking about products price makes it more

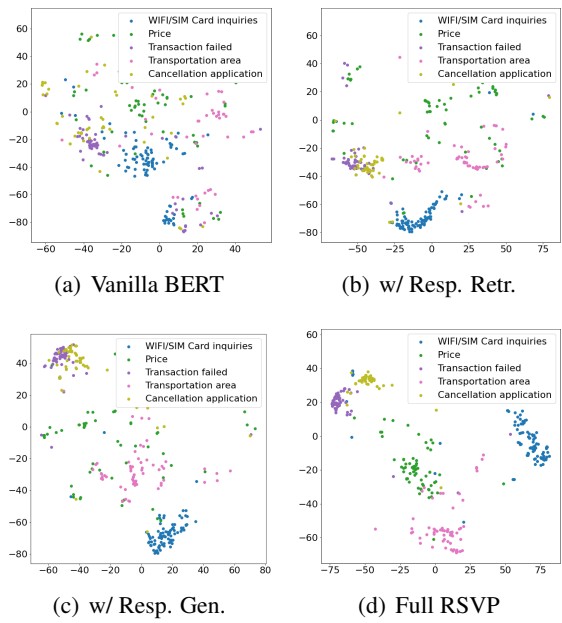

(a) Vanilla BERT      (b) w/ Resp. Retr.

(c) w/ Resp. Gen.      (d) Full RSVP

Figure 5: tSNE plots of encoded utterances of KKday.

difficult to group together with the help of only one pre-taining task, yet it can show clearer boundaries with the integration of both objectives (full RSVP). Another interesting observation is the inseparable bonding of two intents *Transaction failed* and *Cancellation application* since they share pretty similar utterances and responses. The results show that only learning to provide an appropriate response for an utterance is not enough for good classification performance, and a fine-tuning stage can relieve the confusion between similar samples.

## 5 Conclusion

This paper presents RSVP, a novel two-stage framework to address the challenging intent detection problem in real-world task-oriented dialogue systems. RSVP leverages customer service agents' responses to pre-train with customer utterances, offering a refreshing view to utilize the metadata in the dialogue systems. It allows us to explore the implicit intent lying in both utterances and responses, and takes advantage of contrastive learning to improve the model's robustness. RSVP consistently outperforms other strong baseline models on two real-world customer service benchmarks, showcasing the effectiveness of this framework. We believe RSVP can serve flexibly for task-oriented dialogue intent detection problems, and several interesting research directions could be further investigated within this framework, such as more metadata related to the conversation, pre-training tasks inte-

grating the label names with responses, etc.

## Limitations

One limitation of the RSVP framework is that it has not dealt with the noisy labels introduced by the agents' annotation in the KKday dataset. This is likely due to being multi-label implicitly for the dataset, as in the real world the intent of a customer utterance may belong to not only one intent class, while it is classified into merely one class. That makes the annotation inconsistent between utterances. Since we leverage agent responses in pretraining, the framework can be further enhanced by incorporating them into the calibration of intent labels to mitigate the limitation. Another limitation is that we have not considered the out-of-scope (OOS) intents. To address this, we plan to utilize a nearest neighbor algorithm along with the response pre-training task to further discriminate against the OOS utterances and responses in future work.

## Acknowledgements

We thank the anonymous EMNLP reviewers for their insightful comments and feedback. This research was supported in part by KKday.

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

# A  Experimental Setup

Table 5 reports the intents of the four datasets, and Table 6 reports the statistics of the two datasets. Uttr. and Resp. denote the Utterance and Response respectively.

## A.1  Implementation Details

We conducted RSVP experiments with the `bert-base-chinese` and `roberta-base` encoders from (Wolf et al., 2020) as the initial models for two datasets with different languages, respectively. The encoders are used for all models: 768-dimensional Transformer layers with 12 attention layers. We used the pre-trained ConvBERT and DNNC[5] as the initialized models of ConvBERT and DNNC. We adopted the training set during the pre-training stage (i.e., response retrieval and response generation), and set the training epochs to 10 for the two pre-training tasks respectively. We set the temperature $\tau$ to 0.8 and batch size $n = 16$ with a maximum length of 512 tokens due to the hardware constraints. During the fine-tuning stage, we trained our model for 15 epochs from the pre-trained checkpoint with a batch size of 10, and set $\lambda$ to 0.5, making $\mathcal{L}_{ce}$ the main summand in $\mathcal{L}_{ft}$. We used the AdamW (Loshchilov and Hutter, 2017) optimizer during both the pre-training and fine-tuning stages with the learning rate to $2e-5$ and a dropout ratio of 0.1. All the experiments were conducted on an NVIDIA RTX 3090.

---

[5]https://github.com/jianguoz/Few-Shot-Intent-Detection

| Dataset | Intents |
|---|---|
| KKday | Itinerary(within prod page), Itinerary(information not in prod page), UI inquiries, Date doesn't open yet, Date can't be selected, Transaction failed, Booking method, Customer can't find the order, Product comparison, Refund, Coupon, MKT campaign Inquiries, About the Ticket/Prod.(How/when to use), Customization inquiries, Payment & Currency inquiries, How to redeem, Details Inquary(Meeting time/point,how to use,etc.), WIFI/SIM Card inquiries, Multiple accounts, KKday points related rules, Price, Confirm booking details, Transaction procedure, Recommendation, Add/Change booking information, Cancellation application, Transportation area, Complaint, Receipt application, System error, Fraudulent, Pushing order, Complaint MKT Campaign, Resend the voucher, Fully booked (Change the date/Cancellation), Log-in inquiries, Positive feedback, Deactivation |
| TwACS | Baggage, BookFlight, ChangeFlight, CheckIn, CustomerService, FlightDelay, FlightEntertainment, FlightFacility, FlightStaff, Other, RequestFeature, Reward, TerminalFacility, TerminalOperation |
| MultiWOZ 2.2 | FindRestaurant, BookRestaurant, FindAttraction, FindHotel, BookHotel, BookTaxi, FindTrain, BookTrain, FindBus, FindHospital, FindPolice |
| SGD | BuyBusTicket, FindBus, TransferMoney, CheckBalance, GetEvents, GetAvailableTime, AddEvent, FindEvents, BuyEventTickets, GetEventDates, SearchOnewayFlight, SearchRoundtripFlights, ReserveOnewayFlight, ReserveRoundtripFlights, FindApartment, ScheduleVisit, ReserveHotel, SearchHotel, BookHouse, SearchHouse, FindMovies, PlayMovie, GetTimesForMovie, LookupSong, PlaySong, LookupMusic, PlayMedia, GetCarsAvailable, ReserveCar, ReserveRestaurant, FindRestaurants, GetRide, BookAppointment, FindProvider, FindAttractions, GetWeather |

Table 5: Dialog intents of the four datasets.

| Dataset | KKday | TwACS |
|---|---|---|
| #Train | 40,219 | 392 |
| #Valid | 5,029 | 49 |
| #Test | 5,028 | 50 |
| #Intents | 38 | 14 |
| #Max Intent | 22,167 | 107 |
| #Min Intent | 1 | 10 |
| Avg Uttr. Length | 44.48 | 40.35 |
| Avg Resp. Length | 185.84 | 60.78 |

Table 6: Statistics of the two intent detection datasets.