# OpenReview forum: "RSVP: Customer Intent Detection via Agent Response Contrastive and Generative Pre-Training"
_EMNLP/2023/Conference — EMNLP 2023 Findings_

### Official Review · Reviewer_GcM6 · 2023-08-04

**Soundness:** 3

**Excitement:**

3: Ambivalent: It has merits (e.g., it reports state-of-the-art results, the idea is nice), but there are key weaknesses (e.g., it describes incremental work), and it can significantly benefit from another round of revision. However, I won't object to accepting it if my co-reviewers champion it.

**Paper Topic And Main Contributions:**

This paper is about the problem of improving the accuracy of intent recognition using pretraining methods. The approach is to use response retrieval and response generation as pretraining tasks to enhance the accuracy of intent recognition.

**Questions For The Authors:**

L288，pre-trained BERT is used as the decoder, could you explain how to use pre-trained BERT as the decoder?

**Reasons To Accept:**

The approach of using response retrieval and response generation as pretraining tasks does have a certain impact on intent recognition.

**Reasons To Reject:**

This paper lacks innovation, and the improvement in experimental results is not significant.
As for the transferability of the approach, it is unclear whether the pre-trained model would be helpful for intent recognition in other scenarios.

**Reproducibility:**

4: Could mostly reproduce the results, but there may be some variation because of sample variance or minor variations in their interpretation of the protocol or method.

**Reviewer Confidence:**

4: Quite sure. I tried to check the important points carefully. It's unlikely, though conceivable, that I missed something that should affect my ratings.

---

> ### Author Rebuttal · Authors · 2023-08-29
>
> Thank you so much for your helpful comments. We are glad you find our approach of using response in our two pre-training tasks has impacts on intent detection task. Our point-by-point responses to your comments and questions are listed below.
>
> > [R1] This paper lacks innovation, and the improvement in experimental results is not significant. As for the transferability of the approach, it is unclear whether the pre-trained model would be helpful for intent recognition in other scenarios.
>
> We would like to clarify the main novelty of our paper. 1) Our generic pretraining design allows us to utilize the in-house metadata in the dialogue systems instead of using additional large-scale datasets to pre-train the model. 2) RSVP can be swiftly adapted to a variety of domains with task-oriented dialogue scenarios in similar contexts (i.e., utterance and metadata). 3) As stated in the caption of Table 1, our improvements over other baselines are statistically significant with a p-value < 0.05. In addition, we have further included two additional datasets to validate our proposed approach. Experimental results in different language datasets can also support this and the analysis provides useful information (e.g., the effects of $L_{uns\\_cl}$ from L471 and the relations between intent boundaries and utterances in L542) to the community.
>
> To further discuss the transferability and robustness of our approach, we experiment with two public dialogue datasets, namely MultiWOZ 2.2 [1] and the SGD [2] datasets. Both of them are initially designed for intent recognition tasks on each utterance, while we manage to detect the intents of the whole dialogue; therefore, we reframe the customer intent in a dialogue as the set of active intent appearing in all of the utterances for these two datasets, and thus the intent detection task will become a multi-label classification task. We consider the F1 score (f1) and accuracy (acc) as our evaluation metrics, and the quantitative results are listed in the following tables. We note that the accuracy metrics may not be intuitive in multi-label classification tasks, and we consider evaluating it more strictly with subset accuracy – the set of labels predicted for a sample (with a model predicting probability >  threshold 0.5) must exactly match the corresponding set of labels in ground truth. We can see that RSVP outperforms the strongest baseline CPFT across all cases, showcasing the superiority and transferability of our method.
>
> 1. MultiWOZ 2.2 [1]
> |   | f1 | acc |
> |---|---|---|
> | CPFT | 96.73 | 84.50 |
> | RSVP (ours) | 98.27 | 91.04 |
> | **Improvements** | **1.6%** | **7.7%** |
> 2. SGD [2]
> |   | f1 | acc |
> |---|---|---|
> | CPFT | 65.91 | 32.46 |
> | RSVP (Ours) | 72.88 | 45.77 |
> | **Improvements** | **10.6%** | **41.0%** |
>
> > [Q1] L288，pre-trained BERT is used as the decoder, could you explain how to use pre-trained BERT as the decoder?
>
> As stated in L289-294, we have provided instructions on how to adjust pre-trained BERT for initializing a decoder: 1) Modify the original self-attention mechanism to employ causal masking, preventing attending to subsequent tokens during decoding. 2) Add an encoder-decoder cross-attention layer to the decoder. This adjustment approach provides the benefits of using pre-trained checkpoints for seq2seq tasks, leading to saved pre-training time [3]. We also conducted an ablation experiment on different encoder-decoder architectures (see Table 2: “replace BERT with mT5”), confirming that the method introduced in [3] achieved superior performance.
>
> ---
> ### Reference:
>
> [1] Zang, Xiaoxue, et al. "MultiWOZ 2.2: A Dialogue Dataset with Additional Annotation Corrections and State Tracking Baselines." Proceedings of the 2nd Workshop on Natural Language Processing for Conversational AI. 2020.
>
> [2] Rastogi, Abhinav, et al. "Towards scalable multi-domain conversational agents: The schema-guided dialogue dataset." Proceedings of the AAAI conference on artificial intelligence. Vol. 34. No. 05. 2020.
>
> [3] Sascha Rothe, Shashi Narayan, and Aliaksei Severyn. 2020. Leveraging Pre-trained Checkpoints for Sequence Generation Tasks. Transactions of the Association for Computational Linguistics, 8:264–280.

---

### Official Review · Reviewer_KLoz · 2023-08-04

**Soundness:** 3

**Excitement:**

3: Ambivalent: It has merits (e.g., it reports state-of-the-art results, the idea is nice), but there are key weaknesses (e.g., it describes incremental work), and it can significantly benefit from another round of revision. However, I won't object to accepting it if my co-reviewers champion it.

**Paper Topic And Main Contributions:**

The paper proposes to use response generation and retrieval as pre-training tasks for building an intent detection model. The main hypothesis is leveraging agent responses to learn a good pre-trained model for intent detection task in task oriented conversational system.

**Reasons To Accept:**

The paper is generally well written and easy to follow. It also addresses a practical task of intent detection in real-world conversational systems.

Experiments are conducted on two datasets of conversational intent detection on both Chinese and English.

**Reasons To Reject:**

The novelty of the paper is limited compared to the existing body of work of using pre-training tasks to improve model performance on intent detection. The contribution of the paper over the works in Zhang et al. 2021 and Vulić et al. 2021 appear incremental. From Table 1, the improvement over CPFT for the larger KKday dataset also seems marginal.

The paper makes a strong assumption that if the model generates an accurate response it implies that the model has understood the intent. This may not hold true for several real word customer service contacts where the customer mentions an unknown intent, or the initial turns from agents ask clarifying questions to customers.

I would also like to see evaluation on other benchmark dialogue datasets such as SGD and MultiWoZ which contain user-system responses and the active intents in the conversation.

**Reproducibility:**

3: Could reproduce the results with some difficulty. The settings of parameters are underspecified or subjectively determined; the training/evaluation data are not widely available.

**Reviewer Confidence:**

4: Quite sure. I tried to check the important points carefully. It's unlikely, though conceivable, that I missed something that should affect my ratings.

---

> ### Author Rebuttal · Authors · 2023-08-29
>
> We are exceedingly grateful for your insightful comments and really appreciate your time in reading our paper! We are excited that you acknowledge the practical usage of our method in real-world dialogue systems and the experiments on both Chinese and English. Our point-by-point responses to your comments are listed below.
>
> > [R1] The novelty of the paper is limited compared to the existing body of work of using pre-training tasks to improve model performance on intent detection. The contribution of the paper over the works in Zhang et al. 2021 [1] and Vulić et al. 2021 [2] appear incremental. From Table 1, the improvement over CPFT for the larger KKday dataset also seems marginal.
>
> We would like to highlight the differences compared with existing approaches. [2] incorporate an additional Reddit dataset for the pre-training, while we only use the in-house dialogue dataset related to the downstream intent detection task without the need for pre-training on other large-scale datasets. Besides, their response ranking task which uses the multiple negatives ranking loss is also different from our proposed response retrieval and response generation pretraining tasks whose goals are to learn agent knowledge before classifying customers' intents.
>
> While we share some similarities with [1] in terms of the usage of contrastive learning, our approach introduces key differences: (1) The existing work is not able to take advantage of the metadata (agent response), which is beneficial for learning fine-grained conversational text encoder. (2) The supervised contrastive learning loss in their finetuning stage may restrict them to a multi-label context, as it is difficult to define and sample the positive and negative pairs for each multi-intent instance. In contrast, we use unsupervised contrastive learning with the dropout function for augmentation.
>
> We summarize the discussions of experimenting with additional datasets in the response to R3 to reduce repetitive responses.
>
> > [R2] The paper makes a strong assumption that if the model generates an accurate response it implies that the model has understood the intent. This may not hold true for several real word customer service contacts where the customer mentions an unknown intent, or the initial turns from agents ask clarifying questions to customers.
>
> Indeed, the agent response may still be in initial turn and the intent is unknown in practice. Therefore, we incorporate the responses from historical records into the pre-training stage to enable the model to mimic the agent providing a response, which can be viewed as the orientation when a new agent joins the company to learn customers’ intents from previous records. As the agent is able to infer the customer intent before answering the utterance, our model can also be equipped with this ability after the proposed pre-training tasks. During the real-world inference stage, our RSVP with such knowledge can classify intents solely based on the mentioned scenarios without agent responses.
>
> > [R3] I would also like to see evaluation on other benchmark dialogue datasets such as SGD and MultiWoZ which contain user-system responses and the active intents in the conversation.
>
> Thanks for providing these constructive suggestions! We conducted experiments on additional two datasets (MultiWOZ 2.2 [3] and SGD [4]) with RSVP and the strongest baseline CPFT. These two datasets share a similar format and need to detect the intent of each utterance, not the whole dialogue as our settings. To address this issue, we reformulate the customer intent in a dialogue as the set of active intent appearing in all of the utterances for these two datasets.
>
> We use the F1 score (f1) and accuracy (acc) as the evaluation metrics and show the results in the following tables respectively. We note that the accuracy metrics may not be intuitive in multi-label classification tasks, and we consider evaluating it more strictly with subset accuracy – the set of labels predicted for a sample (with a model predicting probability >  threshold 0.5) must exactly match the corresponding set of labels in ground truth. We can see that RSVP can outperform CPFT in all cases, which further supports the robustness of our method. The results are also aligned with our perspective in R1, “CPFT might be restricted to multi-label context due to the SCL loss term”. We thank the reviewer for this and will definitely include these results and analysis!
>
> 1. MultiWOZ 2.2 [3]
> |   | f1 | acc |
> |---|---|---|
> | CPFT | 96.73 | 84.50 |
> | RSVP (ours) | 98.27 | 91.04 |
> | **Improvements** | **1.6%** | **7.7%** |
> 2. SGD [4]
> |   | f1 | acc |
> |---|---|---|
> | CPFT | 65.91 | 32.46 |
> | RSVP (Ours) | 72.88 | 45.77 |
> | **Improvements** | **10.6%** | **41.0%** |
>
>
> ---
> ### Reference:
>
> [1] Zhang, Jianguo, et al. "Few-Shot Intent Detection via Contrastive Pre-Training and Fine-Tuning." Proceedings of the 2021 Conference on Empirical Methods in Natural Language Processing. 2021.
>
> [2] Vulić, Ivan, et al. "ConvFiT: Conversational Fine-Tuning of Pretrained Language Models." Proceedings of the 2021 Conference on Empirical Methods in Natural Language Processing. 2021.
>
> [3] Zang, Xiaoxue, et al. "MultiWOZ 2.2: A Dialogue Dataset with Additional Annotation Corrections and State Tracking Baselines." Proceedings of the 2nd Workshop on Natural Language Processing for Conversational AI. 2020.
>
> [4] Rastogi, Abhinav, et al. "Towards scalable multi-domain conversational agents: The schema-guided dialogue dataset." Proceedings of the AAAI conference on artificial intelligence. Vol. 34. No. 05. 2020.

---

### Official Review · Reviewer_ChMc · 2023-08-11

**Soundness:** 4

**Excitement:**

3: Ambivalent: It has merits (e.g., it reports state-of-the-art results, the idea is nice), but there are key weaknesses (e.g., it describes incremental work), and it can significantly benefit from another round of revision. However, I won't object to accepting it if my co-reviewers champion it.

**Paper Topic And Main Contributions:**

This paper proposes RSVP, a 2-stage pretraining + fine-tuning framework for training intent detectors. Experiments on two customer service datasets show the proposed method achieve superiors performance compared to various baseline methods.

**Questions For The Authors:**

Q1: At section 4.1  mentions that publicly available intent detection benchmarks usually don't come with dialogues, however it seems that there are existing work [1] for intent detection using large-scale publicly available dataset (SGD dataset) with agent responses [2]?

Not that every dataset set need to be included, but I feel it'd be helpful if the authors could consider incorporating at least one larger-scale, publicly available dataset for comparison.

[1] Abhinav Rastogi, Xiaoxue Zang, Srinivas Sunkara, Raghav Gupta, and Pranav Khaitan. AAAI 2019. Towards scalable multi-domain conversational agents: The schema-guided dialogue dataset.
[2] Li Zhang, Qing Lyu, and Chris Callison-Burch. AACL 2020. Intent Detection with WikiHow.

**Reasons To Accept:**

- The proposed method is intuitive and has solid performance and is well supported by it's experiments and evaluations.
- The paper is well written and easy to follow.


**Reasons To Reject:**


See questions


**Reproducibility:**

3: Could reproduce the results with some difficulty. The settings of parameters are underspecified or subjectively determined; the training/evaluation data are not widely available.

**Reviewer Confidence:**

4: Quite sure. I tried to check the important points carefully. It's unlikely, though conceivable, that I missed something that should affect my ratings.

---

> ### Author Rebuttal · Authors · 2023-08-29
>
> We sincerely appreciate your positive review comments. We are glad you find our proposed method intuitive and the performance solid and well supported by the experiments. Our response to the question is given below.
>
> > [Q1.1] At section 4.1 mentions that publicly available intent detection benchmarks usually don't come with dialogues, however it seems that there are existing work [1] for intent detection using large-scale publicly available dataset (SGD dataset) with agent responses [3]?
>
> Thanks for the suggestions! We delve into these mentioned works and find that the Wikihow dataset has a key difference from our scenario, where the response is represented as their intent label. For example, the utterance may be “let check-in agents and flight attendants know if it’s a special occasion”, and the response would be “Get Upgraded to Business Class”; in this case, the response is also the intent label of the utterance. Therefore, the intent label is open-ended and is not a closed set.
>
> > [Q1.2] Not that every dataset set need to be included, but I feel it'd be helpful if the authors could consider incorporating at least one larger-scale, publicly available dataset for comparison.
>
> Thanks for the constructive comments! We add the experiment to the MultiWOZ 2.2 [2] and SGD [3] datasets accordingly and present the results in the following tables. These two datasets were originally designed to detect the active intent of each utterance rather than the whole dialogue as our settings. To provide fair comparisons as ours, we reformulate the customer intent in a dialogue as the set of active intent appearing in all of the utterances for these two datasets (i.e., multi-label classification task).
>
> The F1 score (f1) and accuracy (acc) are used as our evaluation metrics. We note that the accuracy metrics may not be intuitive in multi-label classification tasks, and we consider evaluating it more strictly with subset accuracy – the set of labels predicted for a sample (with a model predicting probability >  threshold 0.5) must exactly match the corresponding set of labels in ground truth. We compare the best-performing baselines (CPFT) with our RSVP and the results are presented in the following tables for two additional datasets. These results are run with 5 seeds. We can observe that RSVP still outperforms CPFT similar to our findings in the paper. We will add the experimental results and discussions to our final version.
>
> 1. MultiWOZ 2.2 [2]
> |   | f1 | acc |
> |---|---|---|
> | CPFT | 96.73 | 84.50 |
> | RSVP (ours) | 98.27 | 91.04 |
> | **Improvements** | **1.6%** | **7.7%** |
> 2. SGD [3]
> |   | f1 | acc |
> |---|---|---|
> | CPFT | 65.91 | 32.46 |
> | RSVP (Ours) | 72.88 | 45.77 |
> | **Improvements** | **10.6%** | **41.0%** |
>
> ---
> ### Reference:
>
> [1] Li Zhang, Qing Lyu, and Chris Callison-Burch. AACL 2020. Intent Detection with WikiHow.
>
> [2] Zang, Xiaoxue, et al. "MultiWOZ 2.2: A Dialogue Dataset with Additional Annotation Corrections and State Tracking Baselines." Proceedings of the 2nd Workshop on Natural Language Processing for Conversational AI. 2020.
>
> [3] Rastogi, Abhinav, et al. "Towards scalable multi-domain conversational agents: The schema-guided dialogue dataset." Proceedings of the AAAI conference on artificial intelligence. 2020.

---

### Official Review · Reviewer_tLuo · 2023-08-12

**Soundness:** 3

**Excitement:**

4: Strong: This paper deepens the understanding of some phenomenon or lowers the barriers to an existing research direction.

**Paper Topic And Main Contributions:**

Existing intent detection methods ignore the use of agent response. In this paper, we propose RSVP, a self-supervised framework dedicated to task-oriented dialogues, which utilizes agent responses for pre-training, which includes response retrieval and response generation task. Experiment demenstrate the effectiveness of the method.

**Questions For The Authors:**

1. Results on more datasets.
2. What will happen when right response doesn't represent the right intent detection?

**Reasons To Accept:**

1. Proposes a novel approach to use agent response and two specific pretraining task for improving the intent detection.
2. Provides both qualitative analysis and quantitative metrics demonstrating this method improve the performance of intent detection.
3. he paper is meticulously structured and articulated, making it exceptionally easy to follow and understand.

**Reasons To Reject:**

1. Only two datasets (with a very small one) are not enough for experiments.
2. Right response doesn't represent the right intent detection. There is no discussion about this.

**Reproducibility:**

4: Could mostly reproduce the results, but there may be some variation because of sample variance or minor variations in their interpretation of the protocol or method.

**Reviewer Confidence:**

4: Quite sure. I tried to check the important points carefully. It's unlikely, though conceivable, that I missed something that should affect my ratings.

---

> ### Author Rebuttal · Authors · 2023-08-29
>
> Thank you very much for your constructive suggestions. We are excited that you recognize our novelty in using agent response with two pre-training tasks and find our paper to be meticulously structured. We have discussed more results with additional datasets and our point-to-point responses to your questions are listed below.
>
> > [R1, Q1] Only two datasets are not enough for experiments. More results on additional datasets.
>
> Thank you for the suggestions! To further illustrate the comparisons between RSVP and the baselines, we experiment with two additional datasets as suggested by Reviewer ChMc and KLoz, MultiWOZ 2.2 [1] and SGD [2], which contain user utterances, agent responses, and user intents in a conversation. These two datasets were originally designed to detect the active intent of each utterance rather than the whole dialogue as our settings. To provide fair comparisons as ours, we reformulate the customer intent in a dialogue as the set of active intent appearing in all of the utterances for these two datasets (i.e., multi-label classification task).
>
> The F1 score (f1) and accuracy (acc) are used as our evaluation metrics. We note that the accuracy metrics may not be intuitive in multi-label classification tasks, and thus we evaluate it more strictly with subset accuracy – the set of labels predicted for a sample (with a model predicting probability >  threshold 0.5) must exactly match the corresponding set of labels in ground truth. We compare the best-performing baselines (CPFT) with our RSVP and the results are presented in the following tables for two additional datasets. These results are run with 5 seeds. It can be observed that RSVP outperforms CPFT in all cases, which also validates the claims in our paper. We will add these results to the final version.
>
> 1. MultiWOZ 2.2 [1]
> |   | f1 | acc |
> |---|---|---|
> | CPFT | 96.73 | 84.50 |
> | RSVP (ours) | 98.27 | 91.04 |
> | **Improvements** | **1.6%** | **7.7%** |
> 2. SGD [2]
> |   | f1 | acc |
> |---|---|---|
> | CPFT | 65.91 | 32.46 |
> | RSVP (Ours) | 72.88 | 45.77 |
> | **Improvements** | **10.6%** | **41.0%** |
>
> > [R2, Q2] Right response doesn't represent the right intent detection. What will happen when right response doesn't represent the right intent detection?
>
> Great question. This scenario may not happen for companies since the agents are responsible for resolving questions raised by customers, where these questions are asked based on customers’ intents. Thus, in our opinion, the mentioned scenario will probably happen for self-volunteering answers on social platforms (e.g., Stack Overflow) that may not address the questions instead of companies that aim to address customers’ inquiries to maintain customer engagement.
>
> For this scenario, it is still possible to classify the correct intent by fine-tuning the pre-trained model with the corresponding intents, which can guide the model to change the importance of incorporating the knowledge of agent responses. Nonetheless, our focus is to support real-world customer services for companies.
>
> ---
> ### Reference:
>
> [1] Zang, Xiaoxue, et al. "MultiWOZ 2.2: A Dialogue Dataset with Additional Annotation Corrections and State Tracking Baselines." Proceedings of the 2nd Workshop on Natural Language Processing for Conversational AI. 2020.
>
> [2] Rastogi, Abhinav, et al. "Towards scalable multi-domain conversational agents: The schema-guided dialogue dataset." Proceedings of the AAAI conference on artificial intelligence. 2020.

---

### Official Review · Reviewer_v7zY · 2023-08-12

**Soundness:** 3

**Excitement:**

3: Ambivalent: It has merits (e.g., it reports state-of-the-art results, the idea is nice), but there are key weaknesses (e.g., it describes incremental work), and it can significantly benefit from another round of revision. However, I won't object to accepting it if my co-reviewers champion it.

**Paper Topic And Main Contributions:**

In this paper, the authors propose a two-stage framework (RSVP) for intent detection of customer utterances in a conversational system. The proposed framework consists of two pre-training tasks that use the agent's response along with the customer's dialog to help the model learn the implicit intent. After the pre-training steps, the model is fine-tuned to predict the intent. Experimental results on the KKday and TwACS datasets show that RSVP outperforms baselines. The authors also conducted ablation studies to validate their two-stage framework. Code to reproduce results and TwACS dataset are provided as part of supplementary materials.

**Questions For The Authors:**

- Accuracy on KKday dataset is low compared to TWaCS even though KKday is much larger in size compared to TWaCS. Is there any explanation for that?

- Is there any plans to release KKday dataset?

**Reasons To Accept:**

- Proposed an interesting approach that uses additional metadata like agents response instead of public conversational dataset to pre-train a model
- Results in English (TWaCS) & Chinese (KKday) datasets shows that the proposed approach works on various languages
- Detailed ablation study on various components of the framework and contrastive learning
- Insightful visualization of few selected utterance representations at various stages of training
- Code is provided as supplementary material

**Reasons To Reject:**

- For given examples, benefit or use of explicit intent detection is not articulated well
- KKday dataset is not provided or publicly available and not decribed well in the paper
- Minor improvement in accuracy compared to a classifier  for KKday dataset

**Reproducibility:**

2: Would be hard pressed to reproduce the results. The contribution depends on data that are simply not available outside the author's institution or consortium; not enough details are provided.

**Reviewer Confidence:**

3: Pretty sure, but there's a chance I missed something. Although I have a good feel for this area in general, I did not carefully check the paper's details, e.g., the math, experimental design, or novelty.

---

> ### Author Rebuttal · Authors · 2023-08-29
>
> Thank you so much for your helpful review. We are excited that the reviewer finds our approach using additional metadata instead of public datasets interesting, and recognizes that our results in various languages are detailed. In addition, we are thankful that the reviewer rates our case studies visualization at various stages of training as insightful. Our point-to-point responses to your comments are given below.
>
> > [R1] For given examples, benefit or use of explicit intent detection is not articulated well
>
> Thanks for your feedback! Intent detection methods are developed to consider different urgent importance and dispatching for the corresponding departments. For instance, website-related problems are often assigned to the engineering department, and product-related utterances may need to be solved by the R&D department. It can be quite time-consuming and high-cost for agents to determine the intents, especially for a large number of intent counts. Therefore, automated customer intent detection benefits companies by reducing human efforts in distributing customers’ inquiries and thus improving the efficiency of agent response. We will rephrase the example to the final version.
>
> > [R2, Q2] KKday dataset is not provided or publicly available and not described well in the paper. Is there any plans to release KKday dataset?
>
> We discussed the KKday dataset in Section 4.1 and summarized the intent metadata and statistics (e.g., average lengths of utterances and responses) in Tables 4 and 5 respectively – The dataset consists of multi-turn utterances and responses for the e-commence travel domain, where the intent labels were annotated by humans. While the KKday dataset is confidential and requires to removal of personalized information, we indeed plan to release the dataset afterward to foster the research community. Meanwhile, we experimented with the public dataset – TwACS and additional datasets (MultiWOZ 2.2 [1] and SGD [2]) as suggested by Reviewers ChMc and KLoz for reproducibility. These quantitative results have showcased the effectiveness of our method.
>
> > [R3] Minor improvement in accuracy compared to a classifier for KKday dataset
>
> We would like to highlight the significant performance of our RSVP and the classifier baseline. Quantitatively, RSVP outperforms the classifier baseline by 3.2% for accuracy, 1.5% for MRR@3, and 1.3% for MRR@5 in terms of the KKday dataset. In addition, although adding the proposed contrastive loss (i.e., $L_{uns\\_cl}$) improves the performance of the classifier, RSVP still surpasses the baseline, which also highlights the generic design of incorporating contrastive learning into different models using only the CE loss. Nonetheless, Our improvements over baselines are still statically significant with a p-value < 0.05. These comparisons on the TwACS dataset further demonstrate the significant improvement.
>
> > [Q1] Accuracy on KKday dataset is low compared to TWaCS even though KKday is much larger in size compared to TWaCS. Is there any explanation for that?
>
> The KKday dataset can be considered as the more challenging real-world scenario compared with the TwACS dataset for the following reasons.
> - The KKday dataset has multi-label implicity for each utterance as mentioned in Section Limitations, where the intent of a customer utterance may belong to not only one intent, while it is classified into merely one class. This provides a unique challenge for RSVP and all the other baselines.
> - In addition, the higher number of labels (38 of KKday v.s. 14 of TwACS) also hinders the performance of each model.
> - Nonetheless, as stated in the caption of Table 1, our experimental results still show statistically significant improvements over baselines, leading to the fact that 1.7% accuracy improvements are helpful in practice.
>
> ---
> ### Reference:
>
> [1] Zang, Xiaoxue, et al. "MultiWOZ 2.2: A Dialogue Dataset with Additional Annotation Corrections and State Tracking Baselines." Proceedings of the 2nd Workshop on Natural Language Processing for Conversational AI. 2020.
>
> [2] Rastogi, Abhinav, et al. "Towards scalable multi-domain conversational agents: The schema-guided dialogue dataset." Proceedings of the AAAI conference on artificial intelligence. 2020.

---

### Meta-Review · Area_Chair_TEk4 · 2023-10-04

**Recommendation:** 3

**Metareview:**

The paper uses response generation and builds an intent detection using gold responses as retrieval as pre-training tasks for building an intent detection model. While it's a great paper, results on more datasets would strengthen the argument.

---

### Decision · Program_Chairs · 2023-10-07

**Decision:**

Accept-Findings

**Comment:**

The paper uses response generation and builds an intent detection using gold responses as retrieval as pre-training tasks for building an intent detection model. While it's a great paper, results on more datasets would strengthen the argument.